# Aging and Possible Benefits or Negatives of Lifelong Endurance Running: How Master Male Athletes Differ from Young Athletes and Elderly Sedentary? [note 1]

**DOI:** 10.3390/ijerph192013184

**Published:** 2022-10-13

**Authors:** Matej Vajda, Ľudmila Oreská, Alena Černáčková, Martin Čupka, Veronika Tirpáková, Ján Cvečka, Dušan Hamar, Feliciano Protasi, Nejc Šarabon, Sandra Zampieri, Stefan Löfler, Helmut Kern, Milan Sedliak

**Affiliations:** 1Hamar Institute for Human Performance, Faculty of Physical Education and Sports, Comenius University in Bratislava, 814 69 Bratislava, Slovakia; 2Department of Biological and Medical Sciences, Faculty of Physical Education and Sports, Comenius University in Bratislava, 814 69 Bratislava, Slovakia; 3Institute of Sports Medicine, Faculty of Medicine, Slovak Medical University, 831 01 Bratislava, Slovakia; 4Center for Advanced Studies and Technology (CAST), University G. D’Annunzio of Chieti-Pescara, 66100 Chieti, Italy; 5Faculty of Health Sciences, University of Primorska, SI-6310 Izola, Slovenia; 6Human Health Department, InnoRenew CoE, SI-6310 Izola, Slovenia; 7Laboratory for Motor Control and Motor Behavior, S2P, Science to Practice, Ltd., SI-1000 Ljubljana, Slovenia; 8Department of Biomedical Sciences, University of Padova, 35131 Padova, Italy; 9Department of Surgery, Oncology, and Gastroenterology, University of Padova, 35128 Padova, Italy; 10Ludwig Boltzmann Institute for Rehabilitation Research, 1100 Vienna, Austria; 11Institute for Physical Medicine, Physik und Rheumatherapie, 3100 St. Pölten, Austria; 12Institute of Physical Medicine and Rehabilitation, 3300 Amstetten, Austria

**Keywords:** seniors, endurance runners, physical fitness, muscle aging, body system deterioration

## Abstract

Regular physical activity, recommended by the WHO, is crucial in maintaining a good physical fitness level and health status and slows down the effects of aging. However, there is a lack of knowledge of whether lifelong endurance running, with a volume and frequency above the WHO limits, still brings the same benefits, or several negative effects too. The present study aims to examine the protentional benefits and risks of lifelong endurance running training in Master male athletes, as this level of physical activity is above the WHO recommendations. Within the study, four main groups of participants will be included: (1) endurance-trained master athletes, (2) endurance-trained young athletes, (3) young sedentary adults, and (4) elderly sedentary. Both groups of athletes are strictly marathon runners, who are still actively running. The broad spectrum of the diagnostic tests, from the questionnaires, physical fitness testing, and blood sampling to muscle biopsy, will be performed to obtain the possibility of complexly analyzing the effects of lifelong endurance physical activity on the human body and aging. Moreover, the study will try to discover and explain new relationships between endurance running and diagnostic parameters, not only within aging.

## 1. Introduction

Aging is a natural multisystem process affecting the individual’s body on several levels, interfering with each other. This process is characterized by progressive neural, cardiovascular, respiratory, metabolic, and skeletal muscle systems malfunctioning, manifesting as a decline of the elderly’s physical fitness and functional independence [1,2]. The relationship between human aging and their physiological function is highly individualistic and affected by inactivity [3]. On the other hand, it has been shown that reaching and maintaining a high level of both physical fitness and activity during the life span could be an effective strategy to reduce the age-related decline in physiological functions [4].

For adults (18–64) and older adults (≥65), the World Health Organization’s [5] current recommendation for physical activity for maintaining health and fitness proposes a multimodal program with 150–300 min of moderate and/or 75–150 min vigorous-intensity of endurance activity per week. Additionally, the WHO [5] also suggests increasing the time of moderate and/or vigorous activity over the recommended load for other health benefits. However, the upper limit is not given. Interestingly, there is a group of individuals, consisting of older adults, with an excessive level of physical activity and fitness, generally coined as master athletes.

Some authors suggested that master athletes represent a sample, which may be considered an “across the life span” and could be a feasible model to estimate healthy human aging [6]. In endurance sports, the physical fitness changes are evaluated mainly by cardiorespiratory fitness (CRF) testing. The CRF reflects the overall capacity of the cardiovascular and respiratory systems [7], which is measured, for instance, through maximal or peak oxygen consumption (VO_2_max). CRF could be assessed by various parameters, however, the VO_2_max is considered as a golden standard [8]. There is a linear relationship between the decline in the endurance performance and the Vo2 max in master athletes [9,10]. In addition, VO_2_max is also a predictor of cardiovascular disease and all-cause mortality [11]. The age-related decline in the VO_2_max across the lifespan is reported to be 5–10% per decade and is accelerated to more than 20% at ages over 70 years in untrained individuals [12]. In master endurance athletes, the proportion of decline can be reduced by maintaining the combination of high-intensity and volume of training up to the age of 70 years [13]. As in the general population, a significant acceleration of the decline in the VO_2_max is present in master athletes after reaching the age of 70 occurs [10]. A decrease in the CRF is caused not only by changes at the levels of the central factors such as maximal heart rate, stroke volume, and an A-V O_2_ difference which are well-documented [9], but also changes in the peripheral factors have significant effects on the decreasing CRF.

One of the well-studied and crucial peripheral factors, associated with the decline in CRF even in master athletes in their advanced age, is an age-related alteration in the skeletal muscle, which is likely a multifactorial process with a network of interacting malfunctional systems resulting in the accelerated loss of muscle mass, and a reduction in muscle strength and power [14,15,16]. This process is well known as sarcopenia. Several recent research studies showed that muscle-related alterations might be caused by factors such as central and peripheral nerve dysfunction and variations, the decrease and remodeling of motor units, fat and connective tissue infiltration, and the reduction and remodeling modifications in the cross-sectional muscle area (type II. muscle fibers reduction compared to type I. muscle fibers, and possible reinnervation) [16,17,18,19]. This multifactorial process is associated with complex structural and functional changes on the cellular level of the muscle fibers, such as a reduced satellite cells content, myonuclear number, capillarization, or mitochondrial dysfunction [16,20]. Likewise, on the subcellular and molecular levels, for example, dysfunction in EC coupling, loss and misplacement of mitochondria, and SOCE impairment have all been described within ageing. Altogether these alterations are accompanied by a development of low-grade chronic inflammation (inflammaging) [21,22].

Another factor which affects CRF is malnutrition caused by prolonged inadequate energy and/or protein intake [23,24]. In athletes, a relative energy deficiency is defined as “impaired physiological functioning caused by a relative energy deficiency and includes but is not limited to impairments of metabolic rate, menstrual function, bone health, immunity, protein synthesis, and cardiovascular health” [25]. From the aging perspective, the relative energy deficiency may negatively affect bone health and might increase the risks of osteoporosis development, which might be associated with higher risks of several fractures and injuries in the higher age of both elderly and master athletes [26,27].

The study primarily aims to investigate the benefits, and possible negative effects, of lifelong endurance training in master athletes, who actively run marathons, and compare these with elderly and young counterparts, who are less active than recommended by the WHO [5], as well as with young endurance trained athletes, who also run marathons.

Therefore, based on the evidence, the specific objectives of the study are: To examine the (a) cardiorespiratory fitness, (b) skeletal muscle structural and functional, (c) inflammatory, (d) bone health, and (e) nutritional intake parameters.To determine the effects of aging and the physical activity level on the measured parameters.To describe the relationship between the ageing and physical activity level with monitored parameters.To identify the health benefits, and possible negative effects, of lifelong one-sided endurance training.

## 2. Materials and Methods

### 2.1. Study Design

A descriptive cross-sectional study design is used among the four groups of participants. The study has been planned to be conducted from September 2021 due to gradual realization of some testing procedures.. The study protocol is designed based on SPIRIT Guidelines, and carefully described by using the SPIRIT checklist of items to be included in reports of cross-sectional studies.

### 2.2. Data Collection Scheme

Within the first hiring meeting for the study, the subjects will be fully informed about all procedures’ nature and possible risks before providing written informed consent. The familiarization, physical performance testing, and blood collection will be held at the Diagnostic Center of Prof. Hamar, Faculty of Physical Education and Sports, Comenius University in Bratislava, and performed by experienced examiners and healthcare professionals. DXA and muscle biopsy will be performed by certified medical professionals, surgeon at the University Hospital Ružinov, and University Hospital Antolska in Bratislava, respectively. All data will be assessed at 3 visits, and the time overview of procedures is shown in the Figure 1.

### 2.3. Sample Size

Based on the pilot study, with a mean effect size and a power of 0.9, the sample size of 14 participants per group would be needed to obtain a significant result. However, we will hire 20 subjects per group (in total, 80 subjects) to eliminate potential risks of drop out on the study outcomes. Sample size was calculated using G * Power 3.1.9.2 software.

### 2.4. Subjects

The study will involve a total of 80 healthy male subjects divided into four groups (n = 80): group 1, endurance-trained runner athletes (young athletes—YA; n = 20, age range 20–30 years); group 2, adults less active than recommended (young sedentary—YS; n = 20, age range 20–30 years), group 3, master endurance running athletes (master athletes—MA; n = 20, age range 65–75 years); group 4, seniors less active than recommended (elderly sedentary—ES; n = 20, age range 65–75 years, respectively). The groups of young and master endurance-trained runner athletes will only consist of the active marathon runners.

Subjects must meet the following inclusion criteria to participate in this study:(a)for athletes’ groups:
defined as more than 300 min per week of running activity which is, by ACSM [8], considered as vigorous intensity of endurance activity.Both groups must regularly participate in running competitions (in 10 km, half, and full marathon) for young athletes for at least 3 years and for master athletes for at least 25 years.Have a personal best on 10 km run in last season under 35 min in YA group and under 55 min in MA.(b)for groups less active than recommended: no history of regular physical activity training and no more practice than 150 min of moderate or 75 min of vigorous intensity per week.

The standard inclusion criterion for every group will be body mass index (BMI) (range 18.5–30 kg/m^2^).

Exclusion criteria will be recent or current infection, physical disabilities, malignant disease, cardiovascular, metabolic, autoimmune diseases, malnutrition, and pharmacological interference (e.g., steroids, nonsteroidal anti-inflammatory agents, immunosuppressive and antineoplastic drugs). Moreover, the use of performance-enhancing drugs in the past and during the study period is exclusionary.

### 2.5. Ethics

Permission to conduct the study and to collect the data has already been obtained from the Ethical Committee of the University Hospital Bratislava–Hospital of Ladislav Dérer, the Academician (number 31/2020), following the ethical standards of the Helsinki Declaration and its later amendments. The data collection procedure is explained to all participants before data collection on their voluntary participation and the right to withdraw from the study without any consequences. The study is registered under the Clinical trials gov., registration number: NCT05053282. The full protocol and dataset are available from the corresponding author on reasonable request.

### 2.6. Study Procedures

#### Familiarization

A familiarization session will be held on Monday, 2 days before performance testing, to secure the physical performance tests’ validity. Firstly, subjects will bring signed informed contents and fill in the questionnaires. Afterward, each participant will undergo a medical check consisting of heart rate, blood pressure, resting ECG measurements, and anamnesis realized by a medical doctor. After the medical check, subjects will be familiarized with each testing procedures and equipment.

## 3. The Primary Outcome

The primary outcome measurements will be represented by the VO_2peak_ reached in the cardiorespiratory fitness testing on the cycle ergometer, as these parameters define the physical fitness level for each subject.

### Cardiorespiratory Fitness

Cardiorespiratory fitness will be measured by the direct measurement of the VO_2peak_ through a graded cycling ergometry in a room with a standard temperature and humidity. Due to safety considerations, we decided to use cycling ergometry instead of treadmill walking/running considering the limitations of this procedure [28].

Protocols for cycle ergometry testing will include an initial 3-min warm-up period with a load between 20 and 100 Watts (W) followed by maximal graded exercise with an incremental load between 20 and 30 W/min. The load will be set individually for each subject depending on the predicted work capacity according to the reference values and the participant’s estimated fitness level [8].

The cardiorespiratory variables, VO_2_, carbon dioxide production, respiratory exchange ratio, and pulmonary ventilation will be measured using a gas analyzer Power Cube (Ganshorn Medizin Electronic, Niederlauer, GER) coupled to the computerized system, with samples every 10 s. The gas analyzer of O_2_ and the room environment will be calibrated according to manufacturer’s specification before each test.

The VO_2peak_ will be considered as the highest value obtained during the test. The blood pressure will be monitored before, throughout, and after the test by Metronik BL-6 (STOLL Medizintechnik, Waiblingen, GER), and the heart rate (HR) by 12-lead ECG Quark T12 (Cosmed, Rome, ITA, Rome, Italy). The test will be carried out by an exercise physiologist and a medical doctor.

## 4. Secondary Outcome Measure

### 4.1. Questionnaires

At the familiarization session, subjects will obtain the following questionnaires: (1) Quality of Life Questionnaire (DABQ), (2) Low Energy Availability in Females Questionnaire (LEAF-Q) [29], and (3) The Aging Males Symptoms Scale (AMS) [30]. Currently, there is no similar questionnaire for male athletes as LEAF-Q. Therefore, we decided to use them without questions according to menstruation. The LEAF-Q, along with AMS, will be used to examine the prevalence of RED-S. The questionnaires will be evaluated separately. The questionnaires were translated into the Slovak language according to the WHO [5].

### 4.2. Body Composition and Bone Status

The basic anthropometric parameter measurements such as body height and body weight are the input parameters for measuring body composition by the DXA. Body height will be measured and determined to the nearest centimeter on a digital free-standing stadiometer InBody BSM 170 (InBody Co., Ltd., Cerritos, CA, USA). Body weight will be set, with subjects wearing underwear on Body 230 (InBody CO., Ltd., Cerritos, CA, USA). Afterward, the BMI will be calculated.

The bone mineral density, content, and total and regional body composition will be assessed from a total body DXA scan. The testing procedure of DXA will be realized according to Kralik et al., 2019 [31]. The subjects will be positioned in a supine position in the middle of the densitometry table with their head straight, space between their arms and torso, palms flat on the table, and feet together secured by a strap. A certified radiologist will conduct all DXA procedures.

### 4.3. Muscular Strength

Lower body strength will be measured as a maximal voluntary contraction (MVC) and rate of torque development (RTD) of both isometric extension and flexion on knee dynamometer (ARS dynamometry, S2P Ltd., Ljubljana, Slovenia) [32,33]. For both tests, subjects will perform two warm-up trials with approximately 50% and 80% of maximal effort with a 30-s rest interval [20,31]. Subsequently, three trials will be performed with maximal effort and will be recorded. The subjects will be instructed to push/pull as fast and hard as possible and hold for 5 s against the lever arms [34,35]. The resting interval will be 90 s between the trials and 3 min between the extension and flexion measurement. Monitored parameters will be the peak torque and RTD in four intervals: 0–50, 0–100, 0–150, and 0–200 ms, respectively. Attempts with the highest values will be used for further analyses.

Upper-body strength will be assessed through the isometric MVC handgrip test by Camry Digital Hand Dynamometer (Zhongshan Camry Electronic, Zhongshan, China). Each subject will be in a seated position with the shoulder adducted, the elbow flexed at 90 degrees, and the wrist flexed between 0 and 30 degrees [36]. The subject will perform three trials for each hand with maximal effort. A one-minute rest interval will be between trials and hands. Attempts with the highest values for each hand will be used for further analyses.

### 4.4. The Physical Activity Monitoring Period

The subjects’ daily physical activity will be tracked using an accelerometer AactivPAL4 (PAL Technologies Ltd., Glasgow, Scotland) that is designed to quantify free-living daily activities. ActivPAL uses information about static and dynamic acceleration to (1) distinguish body posture as sitting/lying, standing, and stepping, and (2) estimate energy expenditure expressed as metabolic equivalents—METs [37]. The accelerometer will be worn under clothing on the anterior midline of the right thigh (half the distance between the knee and the hip). After the physical fitness testing session, the researcher will attach the activPAL to the participants’ thigh using flexible waterproof sleeves and waterproof medical grade adhesive dressing (TegadermTM, 3M Co., St. Paul, MN, USA). Subjects will be requested to continuously wear the accelerometer 24 h per day for 12 days [38]. The subjects will be asked to avoid swimming, sauna, or staying in other wet environments. They will only be allowed to shower. If accelerometers were to be removed for legitimate reasons, the subjects will be required to record in the diary all non-wear periods (such as period/reason), as well as times of getting up, falling asleep, or taking naps. Subjects will be requested to wear the ActivPAL for 12 days and perform their daily activities as usual. However, only 7 days (from Monday to Sunday) will be used for further analyses (Figure 1). Additionally, the participants from athletic groups will be requested to have standard light running training on the last day of the tracking period, as long as it is 24 h before muscle biopsy sampling.

### 4.5. Dietary Tracking

Diet records will be tracked within the seven days (Monday to Sunday) while the subjects will be wearing the accelerometers. Subjects will be educated on dietary reporting strategies, such as servings sizes and comparisons of everyday household items. Energy (kilocalorie) and macronutrients (carbohydrate, fat, and protein) data will be retrieved from each participant’s diary and expressed as a daily average for total and relative intakes. Additionally, a timeline of energy and macronutrient intake during the whole day will be created to identify individual differences and deficiencies (frequency of eating, habits, continuity of food, and training). All analyses will be conducted by nutritional software PLANEAT (Planeat Ltd., Bratislava, Slovakia), which has been previously described by Bajer et al. (2019) [39]. A single researcher will analyze the dietary data to avoid variability in the interpretation of these data.

### 4.6. Blood Collection and Analyses

The venous blood collection will be held 13 days after the physical performance testing session (Figure 1) and collected by a nurse at the hospital. The participants will arrive at the hospital by car between 7:30 a.m. and 8 a.m. The subjects will also be instructed to come after overnight fasting. Blood samples will be collected in several tubes. Plasma will be obtained from venous blood collected into K3EDTA tubes and then centrifuged at 4 °C for 10 min at 1000× *g*. After centrifugation, blood plasma will be collected and stored at −80 °C until processing via an enzyme-linked immunosorbent assay (ELISA) to analyze inflammatory markers (IL-1β, IL-6, TNF-α, CRP), malnutrition biomarkers (transthyretin and retinol binding protein), as well as metabolic hormones (e.g., leptin, adiponectin, ghrelin, and thyroid hormones) held by our laboratory.

Additionally, basic biochemical parameters and malnutrition biomarkers, including glucose, urea, creatinine, uric acid, total proteins, albumins, transthyretin, and retinol binding protein, total bilirubin, alkaline phosphatase, aspartate aminotransferase, alanine aminotransferase, γ-glutamyltransferase, total cholesterol, low-density lipoprotein cholesterol, high-density lipoprotein cholesterol, triacylglycerols, and glycated hemoglobin HbA1c, will be analyzed in blood samples by a commercial medical laboratory.

### 4.7. Muscle Sampling and Analyses

The muscle biopsy will be held after the blood collection (Figure 1). The procedure will be performed at the hospital by a medical doctor and nurse. Subjects will be instructed to avoid any strenuous physical activity 48 h before the procedure and come in the fasted state. The biopsy will be performed with the subject in a supine position and under local anesthesia (lidocaine 2%) using a percutaneous muscle biopsy technique using the sterile Bergström needle (Bergström-Stille, Sweden, 5 mm) with manual suction to obtain muscle specimens (approximately 80 mg) from the mid-section of the right m. vastus lateralis [31]. Afterwards, visible connective tissue and fat will be dissected away from samples.

For histology and immunohistochemistry, a bundle of fibers will be mounted in cryomolds with an OCT embedding compound and snap-frozen immediately in liquid nitrogen-cooled isopentane. The samples will be stored at −80 °C for further analyses. Afterwards, the samples will be cut at an 8 μm thickness at −21 °C using a cryostat and mounted on microscope slides. The sections will be stained and analyzed for their muscle fiber morphometry, nuclear number and myonuclear domain, satellite cells, and capillaries.

The muscle samples for the molecular analyses will be snap-frozen only in liquid nitrogen and stored at −80 °C for further analyses. The tissue will be used for gene expression analysis (PCR). Target genes will include inflammatory markers (e.g., IL-1β, IL-6, TNFα, COX-2) and markers of muscle differentiation and atrophy (e.g., Atrogin-1, MuRF-1, FoxO, MyoD).

For transmission electron microscopy (EM), the muscle sample of each subject will be fixed in freshly prepared 3.5% glutaraldehyde in 0.1 M cacodylate (NaCaCO) buffer, pH 7.4. The fixed bundles of fibers will be stored at 4 °C, until embedding. Afterwards, quantitative analyses of calcium release units (CRUs or triads), mitochondria, and calcium release units (CEUs) will be performed. The following parameters will be quantified: (1) number of CRUs/area, (2) number of mitochondria and mitochondria-CRUs pairs/area [40,41], and (4) CEUs which are formed by SR-stacks (a) and elongated T-tubules at the I band (b). They will be quantified as follows: (a.) percentage of fibers presenting SR-stacks and number of SR-stacks/(per) 100 μm^2^, (b.) we will determine both: (i) the extension of the SR in close association with the T-tubule in 100 μm^2^ of section, and (ii) the total extension of the T-tubule network in 100 μm^2^ of section.

### 4.8. Data Analysis and Management

All collected data will be cleaned, coded, and entered into the computer. The anonymized data will be uploaded to a cloud server for storage. Access to this data store will be password protected. The data collected from the study will be stored on faculty One-drive cloud servers. After data collection, data from the entire study will be retrieved from the One-drive and downloaded as an Excel file (Microsoft Office, Redmond, WA, USA). Data analysis will be performed using the SPSS 23 (IBM, New York, NY, USA).

The descriptive parameters will be reported as mean values ± standard deviations (SD) and 95% of the confidence interval (95% CI). Data will be assessed for normality and homogeneity to ensure that the analysis’ assumptions will be met, using the Shapiro–Wilk test, Levene’s test, visual check of histograms, and Q–Q plots.

A one-way ANOVA with Bonferroni post hoc correction will be used to determine differences between the groups in all measured parameters. Additionally, Cohen’s d effect size (ES) will be used to calculate between-group differences and will be interpreted using the following thresholds: 0–0.2 was trivial, 0.2–0.6 small, 0.6–1.2 moderate, 1.2–2.0 large, 2.0–4.0 very large, and >4.0 extremely large effect [42].

Association analyses will be performed using the Pearson product–moment correlation coefficient. The strength of the relationship was determined using the following criteria: 0.1, small; 0.3, moderate; 0.5, large; 0.7, very large; 0.9, nearly perfect > 0.9 [43].

The results will be considered significant for values of <0.05.

## 5. Discussion

Physical fitness and the level of physical activity play an important role in maintaining health and reducing the negative effects of aging on the human body. Most of the positive effects are due to the implementation of the volume and intensity of physical activities recommended by the WHO [5]. However, the WHO recommendations suggest that increasing the training volume and intensity beyond the recommended range may increase the positive effects on the body and also reduce the negative impact of aging. Nonetheless, no upper limit has been exactly set, and limited numbers of studies have been done to uncover the positive and negative aspects of the volume of preferred physical activity, especially in older adults. On the contrary, a combination of a high volume and intensity of physical activity may possibly not only have health benefits, but also may have a negative impact on body composition, the musculoskeletal system, and others [14,44].

Consequently, the study might have several practical implications for coaches, both master, and young marathon runners, as well as for clinical practitioners and dietitians. The natural decline in the endurance exercise performance of master runners is present due to a lower training volume as well as age-related physiological changes compared to their younger counterparts. However, these changes have been so far poorly explained in the literature from the point of view of the complexity and interactions of human body systems on different levels. Thus, the study may bring new insides and explanations. Furthermore, to our knowledge, relative energetic deficiency is common among endurance runners. Thus, the investigations of the RED syndrome and its effect on the physiological aspect and exercise performance, not only in young endurance runners, but throughout the lifespan of master runners, might be valuable, especially for medical practitioners, and clinical and sports dietitians.

## 6. Conclusions

In the present study, we will focus on revealing the positive effects and potential negative risks associated with lifelong physical activity beyond and under the recommendations defined by WHO. By using a wide range of diagnostic procedures, we gain a unique opportunity to comprehensively assess the impact of lifelong physical activity on the human body and contribute to expanding the knowledge base in the field. In addition to evaluating the current state, we assume the discovery of several relationships between diagnosed parameters, which will allow us to explain and describe the mechanisms responsible for changes occurring in the human body.

## Figures and Tables

**Figure 1 ijerph-19-13184-f001:**
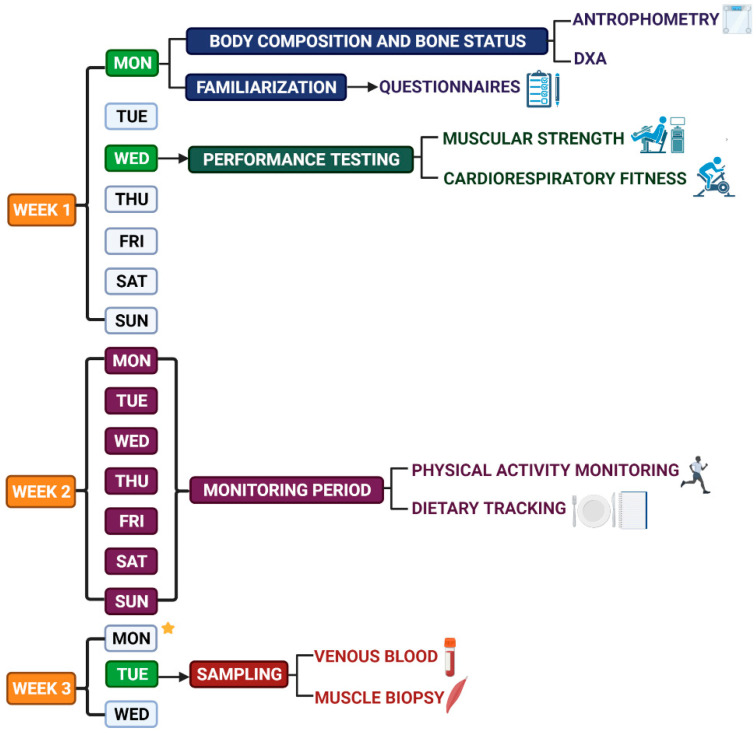
Data Collection Scheme per subject. Created with BioRender.com.

## Data Availability

The datasets analyzed during the current study are available from the corresponding author upon reasonable request.

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
