# Peer review of "Aging and Possible Benefits or Negatives of Lifelong Endurance Running: How Master Male Athletes Differ from Young Athletes and Elderly Sedentary?†"

_ijerph, 2022, doi:10.3390/ijerph192013184_

Round 1
Reviewer 1 Report
This method covers a topic related to public health. The study would focus on important health issues.
Here are my major comments:
The 4 groups selected for comparison follow a consistent logic. However, for practical purposes, a group of older runners who can no longer run should also be included. Indeed, among people who run all their lives, there are some who get injured or have chronic injuries (e.g., body composition disorder, musculoskeletal system problem). Only elderly runners who are still running are included in the 4 groups in this method, and therefore only healthy people, would be included in the protocol, which could be considered as a bias.
How would the authors study the negative effects of lifelong endurance training (it is the main purpose of the study), without a group of lifelong endurance runners who cannot run anymore since they are for instance injured? Otherwise, it is likely that the authors will only highlight the positive effects (or no effect) of lifelong endurance training.
Minor comments:
1. Introduction:
l. 48: Should we understand physiological or cardiorespiratory? In fact, physiology includes anatomical and articular functions. If exercise would have a beneficial effect on the condition of the joints, a reference should be added. Otherwise, the term physiological should be changed to cardiorespiratory.
l. 63: the common abbreviation for peak oxygen consumption is VO2peak and not VO2max.
l. 76-100: The main beneficial effects of physical activity in the fight against the deleterious effects of aging are well listed (e.g., sarcopenia, nutrition). However, the reader would like more detail on the amount of exercise (volume and/or intensity) and age of the population for the studies cited. This is important, as the proposed protocol focuses on the amount (volume/intensity) of physical activity (e.g., running) in different age/training level groups.
1.1. Aims and objectives
It is not clear whether the aim of this study is to include lifelong endurance trained master athletes (i.e., runners, trail race runners, biking, triathletes) or only runners. This should be specified to be consistent with the title.
I would suggest including a couple of basic anthropometric parameters (not only BMI) and a psychological quality of life assessment (if not too late, regarding the overall timeline of the study).
How would the authors study the negative effects of lifelong endurance training, without a group of lifelong endurance runners who cannot run anymore since they are for instance injured? Otherwise, it is likely that the authors will only highlight the positive effects (or no effect) of lifelong endurance training.
2. Methods
l. 158 and throughout the ms: blank space between ‘’10’’ and ‘’km’’
l. 181 and throughout the ms: prefer VO2peak to VO2max, especially for the ES group.
l. 189. the highest value obtained during the test is VO2peak, not VO2max. To achieve VO2max, a plateau needs to be maintained at least 30 s with an increase of VO2 less than 2 ml/kg/min. See Billat and al. publications.
l. 228-238: please describe the method to calculate RTD, especially for this protocol paper. If any software is used (e.g., Labhart, Matlab, or SPSS as described below), please specify it.
l. 294-324: is it the same experimenter who performed the biopsy?
l.325-334: which software was used to assess the statistic power?
l.334. a ‘’p’’ is missing.
Author Response
Review 1
This method covers a topic related to public health. The study would focus on important health issues.
Here are my major comments:
The 4 groups selected for comparison follow a consistent logic. However, for practical purposes, a group of older runners who can no longer run should also be included. Indeed, among people who run all their lives, there are some who get injured or have chronic injuries (e.g., body composition disorder, musculoskeletal system problem). Only elderly runners who are still running are included in the 4 groups in this method, and therefore only healthy people, would be included in the protocol, which could be considered as a bias.
How would the authors study the negative effects of lifelong endurance training (it is the main purpose of the study), without a group of lifelong endurance runners who cannot run anymore since they are for instance injured? Otherwise, it is likely that the authors will only highlight the positive effects (or no effect) of lifelong endurance training.
Dear Reviewer, firstly we would like to say thank for your revision and helpfull comments. We try to incorporate as many of your suggestions as we can to imporove our manuscript. See the commnets and changes below.
Thank you for this comment.
In our work, we focus on a comprehensive assessment of the state of the organism of subjects from 4 selected groups. We focus on two main lines, physical fitness (and it related parameters) and health status. We try to find the relations between parameters of both these lines in each group and then compare differences between selected groups. Moreover, we try to find the relationship between the parameters of these two lines and physical activity level (highly active subjects and active less than recommended).
To maintain levels of both lines (physical fitness and health status) is important to perform regular physical activity. Based on previous studies, there are conceived recommendations from the WHO, ACSM, and others on the recommended level of volume/intensity of movement activity, which have positive effects on both of our monitored lines. What happens to the body when the physical activity is one-side (only running) and the combination of the volume/intensity is high than recommended is not well known, especially in Master athletes who are still active in competing.
When we create study protocol, we come out of our practice, since we have ambulance of a medical doctors in sports, where not only young athletes but also master athletes senior age 60+, who regularly compete at national and international events in endurance sports. These athletes are most often affected by musculoskeletal limitations and injury/diseases and cardiovascular diseases.
Of the musculoskeletal is, for example, tendinopathies and osteoarthritis of the knee and hip joints or deficiency of muscle strength of certain muscle groups. These diseases are considered to be overuse injuries and, in the case of endurance athletes, the incidence and extent of these diseases, in addition to excessive physical activity, the effect may also be limitation of joints mobility, weakening of selected muscle groups and/or lateral deficiency of the lower limbs. Last but not least, nutrition also affects the quality and quantity of bone and muscle mass, and the occurrence of relative energy deficiency and/or low protein intake can have a negative effect on the current condition of the musculoskeletal system. Therefore, in our work, there are parameters that monitor food intake, blood parameters evaluating their settlement in the body as well as the diagnosis of RED syndrome, which occurs in endurance sports. In addition to the above-mentioned musculoskeletal injuries/diseases, there are also diseases of the cardiovascular system such as cardiac remodeling, high blood pressure, atrial fibrillation, and low resting heart rate.
It is important to note that these master athletes are still physically active and regularly compete despite their health problems and limitations. With this information, we can expect even in our study there will be master athletes who will be affected by the disease/injuries and not only healthy individuals.
It also indicates the need for a comprehensive examination of the organism so that we can monitor the complex system of processes that affect each other.
Minor comments:
- Introduction:
- 48: Should we understand physiological or cardiorespiratory? In fact, physiology includes anatomical and articular functions. If exercise would have a beneficial effect on the condition of the joints, a reference should be added. Otherwise, the term physiological should be changed to cardiorespiratory.
It has been changed in the text accordingly to your suggestion.
- 63: the common abbreviation for peak oxygen consumption is VO2peak and not VO2max.
It has been changed in the text accordingly to your suggestion.
- 76-100: The main beneficial effects of physical activity in the fight against the deleterious effects of aging are well listed (e.g., sarcopenia, nutrition). However, the reader would like more detail on the amount of exercise (volume and/or intensity) and age of the population for the studies cited. This is important, as the proposed protocol focuses on the amount (volume/intensity) of physical activity (e.g., running) in different age/training level groups.
Thank for this comment. In general, master athletes are specified as athletes of age over 40 years old, however, in the present study, we are focused to analyse the age category over 60 years old. We used predominantly sources that bring information to this age group.
We add information about age in this sentence: “One of the well-studied and crucial peripheral factors, associated with the decline in CRF even in master athletes (over 60 years old) in their advanced age, is an age-related alteration in the skeletal muscle, which is likely a multifactorial process with a network of interacting malfunction systems resulting to accelerated loss of muscle mass, and reduction of muscle strength, and power [14-16].
About volume, and/or intensity studies used in this section are mostly focused on training status (master athletes, sedentary seniors and etc.) and/or level of cardiorespiratory fitness (Vo2Max/Vo2Peak). We try to find studies that describe training program characteristics but there are limited numbers of studies that are oriented toward Master athletes and they do not present the training characteristics of the subjects.
1.1. Aims and objectives
It is not clear whether the aim of this study is to include lifelong endurance trained master athletes (i.e., runners, trail race runners, biking, triathletes) or only runners. This should be specified to be consistent with the title.
Thank you for your suggestion. Both young and master endurance runners are strictly marathon runners, who must still actively run and attend marathon events. The group of young endurance runners consists of the best Slovak young half marathon and marathon runners. Master endurance athletes have been chosen similarly. According to your suggestion, as well as for easier reading, we added this information to the text. Please, see the changes in the text below:
ABSTRACT: Both groups of athletes are strictly marathon runners, who are still actively running.
The main aim of the study is to investigate the benefits, and possible negative effects of lifelong endurance training in master athletes, who actively run marathons, and compare with elderly and young counterparts, less active than recommended by WHO [5], as well as with young endurance-trained athletes, who also run marathons.
SUBJECTS: The groups of young and master endurance-trained runners athletes will only consist of the active marathon runners.
I would suggest including a couple of basic anthropometric parameters (not only BMI) and a psychological quality of life assessment (if not too late, regarding the overall timeline of the study).
We appreciate your suggestion, and we added to the article the following basic anthropometric parameters, which will be examined, as stated below:
“Body height will be measured and determined to the nearest centimeter on a digital free-standing stadiometer InBody BSM 170 (InBody CO., Lt., Cerritos, USA). Body weight, % of body fat, and lean body mass will be set, with subjects wearing underwear on Body 230 (InBody CO., Lt., Cerritos, USA). Afterward, BMI will be calculated.”
The general quality of life is primarily assessed by the Quality of Life Questionnaire (DABQ). The questionnaire is focused on domains such as socio-demography, lifestyle, health-related questions that includes health limitations during daily activities, health conditions and diseases, feelings and mental diseases, and a comparison of health condition before and after the intervention. Indirectly we assess the quality of life through LEAF Questionnaire and Aging Males Symptoms Scale. The results from the questionnaires will be correlated. However, as you have suggested, an additional assessment of the psychological quality of life will be suitable. We will highly reconsider your suggestion and try to implement it into the ongoing research.
How would the authors study the negative effects of lifelong endurance training, without a group of lifelong endurance runners who cannot run anymore since they are for instance injured? Otherwise, it is likely that the authors will only highlight the positive effects (or no effect) of lifelong endurance training.
This question has been answered early in this response.
- Methods
- 158 and throughout the ms: blank space between ‘’10’’ and ‘’km’’
It has been changed in the text accordingly to your suggestion.
- 181 and throughout the ms: prefer VO2peak to VO2max, especially for the ES group.
It has been changed in the text accordingly to your suggestion.
- 189. the highest value obtained during the test is VO2peak, not VO2max. To achieve VO2max, a plateau needs to be maintained at least 30 s with an increase of VO2 less than 2 ml/kg/min. See Billat and al. publications.
It has been changed in the text accordingly to your suggestion.
- 228-238: please describe the method to calculate RTD, especially for this protocol paper. If any software is used (e.g., Labhart, Matlab, or SPSS as described below), please specify it.
We appreciate your suggestion. We used isometric extension and flexion on a knee dynamometer from ARS dynamometry, S2P Ltd, Ljubljana, Slovenia. This device has software to analyze maximal torque and rate of torque development in the different time periods (0-50,0-100 and etc.). We regularly use outputs from this software for further analyses in our previous publication. We have also two citations of our team members Sarabon et al. 2020 and Bily W et al. 2016 accordingly to these procedures.
- 294-324: is it the same experimenter who performed the biopsy?
The drawing of the biopsy is performed by a doctor and a nurse. The post-biopsy process will be performed by experienced scientists who are fully familiar with all procedures outlined in the manuscript.
l.325-334: which software was used to assess the statistic power?
We used G*Power 3.1.9.2 software.
“Based on the pilot study, with a mean effect size and a power of 0.9, the sample size of 14 participants per group would be needed to obtain a significant result. However, we will hire 20 subjects per group (in total, 80 subjects) to eliminate potential risks of drop out on the study outcomes. The sample size was calculated using G*Power 3.1.9.2 software.”
l.334. a ‘’p’’ is missing.
p values in mentioned at the end of this paragraph.
“The results will be considered significant for values of <0.05.”
Reviewer 2 Report
The main aim of the study „Aging and possible benefits or negatives of the lifelong endurance running: How master male athletes differ from young athletes and elderly sedentary?“ is to investigate the benefits, and possible negative effects of lifelong endurance training in master athletes and compare with elderly and young counterparts, less active than recommended by WHO [5], as well as with young endurance trained athletes.
The study design is interesting. I wish the authors a successful execution of the research. However, some little things need to correct:
Line 101: If there is no part 1.2, there is no reason for 1.1.
Line 164: If there is no part 2.6.2, there is no reason for 2.6.1.
Line 191-192: What indicators will be used to determine the individual initial load?
Author Response
Review 2
The main aim of the study „Aging and possible benefits or negatives of the lifelong endurance running: How master male athletes differ from young athletes and elderly sedentary?“ is to investigate the benefits, and possible negative effects of lifelong endurance training in master athletes and compare with elderly and young counterparts, less active than recommended by WHO [5], as well as with young endurance trained athletes.
Dear Reviewer, firstly we would like to say thank you for your revision and helpful comments. We try to incorporate as many of your suggestions as we can to improve our manuscript. See the comments and changes below.
The study design is interesting. I wish the authors a successful execution of the research. However, some little things need to correct:
Line 101: If there is no part 1.2, there is no reason for 1.1.
Thank for this comment. We decided to not use the numbering system.
Line 164: If there is no part 2.6.2, there is no reason for 2.6.1.
Thank for this comment. We decided to not use a numbering system.
Line 191-192: What indicators will be used to determine the individual initial load?
Thank for this comment. We will use body weight. The range is 20-100 watts - the plan is to use an initial load of around 0.75w/kg in the athlete groups and 0.25w/kg in the sedentary groups.
Reviewer 3 Report
This study aims to examine effects of lifelong endurance training on different aspects of human wellbeing. Although very intriguing, there are lot of questionable parts in this paper. Although, Some aspects of the Introduction and Materials and methods can be significantly improved. Please see some of the comments and raised issues below.
General comments:
Introduction:
Is there any indication about negative consequences of physical activity, especially vigorous one?
There are lack of previous investigations directly related to the topic. The authors stated only theoretical, general assumptions related to the aging process, but there is no data directly related to the impact of training on the aging process.
Chapter Aims and objectives seems like aa part of a master's or doctoral thesis. Is it possible to reconfigure it?
Materials and Methods:
Study design:
I don’t thnik that readers are interested in Overall Study Design Timeline. I think this part is unnecessary.
At Page 4, Line 140 authors said Based on the pilot study…. If there was a pilot study, is it possible to present some results of that pilot study?
Subjects:
I am not fully sure that this group distribution can meet study objectives. Why are younger groups necessary? Why aren't there two middle-aged groups, because that's the period when physical activity decreases the most?
Could the authors explain why they decided on the BMI inclusion criterion range of 18.5-30kg/m2?
Secondary outcome measures – muscular strength:
Are the authors sure that 90 sec rest period between trial attempts is enough?
Discussion:
Authors have to highlight practical implications of the paper (why do authors think that this study is crucial?)
Specific comments:
Page 2, Line 52: Please indicate per week (example 150-300 minutes per week).
Page 2, Line 65: Change VO2max to VO2max
Page 2, Line 73: Did you mean maximal heart rate instead of maximal heart?
Page 2, Lines 76-79: Is it possible to simplify this sentence?
Page 5, Lines 156-157: Is there any reference about ACSM guidelines?
Page 6, Line 211: What is RED-S? Why is that prevalence important?
Author Response
Review 3
his study aims to examine effects of lifelong endurance training on different aspects of human wellbeing. Although very intriguing, there are lot of questionable parts in this paper. Although, Some aspects of the Introduction and Materials and methods can be significantly improved. Please see some of the comments and raised issues below.
General comments:
Introduction:
Is there any indication about negative consequences of physical activity, especially vigorous one?
Dear Reviewer, firstly we would like to say thank you for your revision and helpful comments. We try to incorporate as many of your suggestions as we can to improve our manuscript. See the comments and changes below.
When we create study protocol, we come out of our practice, since we have ambulance of a medical doctor in sports, where not only young athletes but also master athletes senior age 60+, who regularly compete at national and international events in endurance sports. These athletes are most often affected by musculoskeletal limitations and injury/diseases and cardiovascular diseases.
Of the musculoskeletal is, for example, tendinopathies and osteoarthritis of the knee and hip joints or deficiency of muscle strength of certain muscle groups. These diseases are considered to be overuse injuries and, in the case of endurance athletes, the incidence and extent of these diseases, in addition to excessive physical activity, the effect may also be the limitation of joints mobility, weakening of selected muscle groups, or lateral deficiency of the lower limbs. Last but not least, nutrition also affects the quality and quantity of bone and muscle mass, and the occurrence of relative energy deficiency and/or low protein intake can have a negative effect on the current condition of musculoskeletal. Therefore, in our work, there are parameters that monitor food intake, blood parameters evaluating their settlement in the body as well as the diagnosis of RED syndrome, which occurs in endurance sports. In addition to the above-mentioned musculoskeletal injuries/diseases, there are also diseases of the cardiovascular system such as cardiac remodeling, high blood pressure, atrial fibrillation, and low resting heart rate.
It is important to note that these master athletes are still physically active and regularly compete despite their health problems and limitations. With this information, we can expect even in our study there will be master athletes who will be affected by the disease and not only healthy individuals. It also indicates the need for a comprehensive examination of the organism so that we can monitor the complex system of processes that affect each other.
There is lack of previous investigations directly related to the topic. The authors stated only theoretical, general assumptions related to the aging process, but there is no data directly related to the impact of training on the aging process.
Thank for this comment. We try to avoid direct description of data because we would like to use a complex system of processes with huge numbers of parameters and areas. We shortly described the multifactorial impact of aging on the body and malfunction of body systems.
Chapter Aims and objectives seems like aa part of a master's or doctoral thesis. Is it possible to reconfigure it?
Dear reviewer, we reconfigured the Chapter Aims and objectives according to your suggestion.
Materials and Methods:
Study design:
I don’t think that readers are interested in the Overall Study Design Timeline. I think this part is unnecessary.
Dear reviewer, thank you for your suggestion, figure 2. Overall Study Design is removed.
At Page 4, Line 140 authors said Based on the pilot study…. If there was a pilot study, is it possible to present some results of that pilot study?
Dear reviewer, we apologize, but the authors of the pilot study do not wish to present data before the official publishing. If you intend to see some results, they may be requested from the authors.
Subjects:
I am not fully sure that this group distribution can meet study objectives. Why are younger groups necessary? Why aren't there two middle-aged groups, because that's the period when physical activity decreases the most?
We hire 4 groups of subjects, In the sedentary category (young 20 - 30y and seniors 65-75y) are subjects who are not physically active (less than 30 min of regular physical activity weekly). The trend in the European region is that young people are less active than 10 years ago, and more than 30% are classified as sedentary. In our complex analyses would like to identify differences between these 4 groups and we think that master senior athletes can potentially have better physical fitness and some parameters of health status than non-active young adults.
Could the authors explain why they decided on the BMI inclusion criterion range of 18.5-30kg/m2?
Dear reviewer, it is a fine question. Within the study, we are going to choose individuals in the range of BMI from a healthy weight to overweight. The reason, why we decided to include also overweight individuals according to BMI is that elderly are naturally slightly overweight.
The range of BMI from 23.0-29.9kg/m2 is considered normal weight in the elderly according to geriatric BMI calculators. Additionally, in our population, young sedentary adults are usually slightly overweight. The point why we have decided on this wide range is that both young and master endurance-trained athletes are, according to BMI, in the range of healthy weight, on the other side not every sedentary elderly is in this range, and so we try to create their young counterparts among the young sedentary adults.
However, our primary aim is to create as homogenous study groups as possible. We are aware that some overweight individuals might present some hormonal and inflammatory alterations. However, these disturbances are not as significant as in the case of obese individuals. Additionally, each participant, in the range of 25-30.0kg/m2, will be strictly examined for any alterations and disturbances in their metabolic health.
Secondary outcome measures – muscular strength:
Are the authors sure that 90 sec rest period between trial attempts is enough?
We used the same rest period as our team members previously used and published (Sarabon et al. 2020 and Kralik et al. 2019) with a similar protocol on the same device.
Discussion:
Authors have to highlight practical implications of the paper (why do authors think that this study is crucial?).
Dear reviewer, we hope the study may have several practical implications:
- holistic approach in the examination of the age-related changes in master athletes, which of their age-related processes are slowed down and different comparing their elderly counterparts
- how the age-related changes affect the overall exercise performance of master endurance runners comparing their young running counterparts
- how the master runners and young runners differ from their sedentary counterparts in the point of chronobiology, as long as there is a lack of knowledge in the research field
- to examine the effect of lifelong relative energetic deficiency on the various physiological aspects and overall health
- besides the benefits to pinpoint the negative aspect of performing endurance running as the only type of exercise throughout the individual's lifespan
Specific comments:
Page 2, Line 52: Please indicate per week (example 150-300 minutes per week).
It has been changed in the text accordingly to your suggestion.
“150 - 300 minutes of moderate and/or 75 - 150 minutes vigorous-intensity of endurance activity per week”
Page 2, Line 65: Change VO2max to VO2max
It has been changed in the text accordingly to your suggestion.
Page 2, Line 73: Did you mean maximal heart rate instead of maximal heart?
It has been changed in the text accordingly to your suggestion.
Page 2, Lines 76-79: Is it possible to simplify this sentence?
We think this sentence is well written and we would like to use the original version.
Page 5, Lines 156-157: Is there any reference about ACSM guidelines?
Thank for this comment. We previously used ACSM reference in text with the number 8.
“defined as more than 300 minutes per week of running activity which is by ACSM [8] considered as vigorous intensity of endurance activity. “
Page 6, Line 211: What is RED-S? Why is that prevalence important?
Relative energy deficiency in sport (RED-S) describes a syndrome of poor health and declining athletic performance that happens when athletes do not get enough fuel through food to support the energy demands of their daily lives and training. Athletes who suffer from long-term low EA may develop nutrient deficiencies (including anemia), chronic fatigue and an increased risk of infections and illnesses, all of which have the potential to harm health and performance. RED-S can and does affect athletes of any gender and ability level. It has complex health and performance consequences.
More detailed information you can find here: https://bjsm.bmj.com/content/48/7/491?hootPostID=0116e43013bf35a19d2c4f30c76050b1
Round 2
Reviewer 3 Report
Dear authors,
Thank you for answering the questions and thus significantly improving the paper. I hope the questions asked helped you in your study protocol.